# Brief communication - Impact Forecasting Could Substantially Improve the Emergency Management of Deadly Floods: Case Study July 2021 floods in Germany

Heiko Apel[1], Sergiy Vorogushyn[1], and Bruno Merz[1,2]

[1]GFZ German Research Centre for Geoscience, Section Hydrology, Potsdam, Germany
[2]University of Potsdam, Institute for Environmental Sciences and Geography, Potsdam, Germany

*Correspondence to*: Heiko Apel (heiko.apel@gfz-potsdam.de)

**Abstract.** Floods affect more people than any other natural hazard, thus flood warning and disaster management are of utmost importance. However, the operational hydrological forecasts do not provide information about affected areas and impact, but only discharge and water levels at gauges. We show that a simple hydrodynamic model operating with readily available data is able to provide highly localized information on the expected flood extent and impacts, with simulation times enabling operational flood warning. We demonstrate that such an impact forecast would have indicated the deadly potential of the 2021 flood in West Germany with sufficient lead time.

## 1 Introduction

River flooding directly affect, on average, 125 million people annually, by evacuation, homelessness, injury or death (Douben, 2006), and flood exposure and losses are projected to increase owing to climate change and population and socio-economic growth (Dottori et al., 2018). Forecasting and early warning are essential cornerstones of disaster risk reduction as anchored in the Sendai Framework for Disaster Risk Reduction (UNDRR, 2019). However, the official and legally binding operational river flood forecasts in Germany, operated be the different federal states, provide only expected water levels or discharges at specific river gauges. The same holds true for the Global Flood Awareness System GloFAS, developed by the European Commission and the European Centre for Medium-Range Weather Forecasts (ECMWF). The European Flood Alert System EFAS[1] provides warnings with spatial information, but these forecasts have a rather coarse spatial resolution of 100 m, do not consider dikes or other flood protection measures, and are based on pre-calculated hazard maps, not actual flood dynamics. Local decision-makers, disaster managers and potentially affected citizens need more detailed flood information for emergency management decisions. Examples are the decision to issue a disaster alert, to evacuate an urban area, to strengthen levees that may breach, to protect most critical infrastructure objects or to allocate emergency resources to expected damage hotspots. Potentially affected people need to know whether there may be danger to their health and lives, whether their houses and assets

---

[1] https://www.efas.eu

may be at risk of flooding or even destruction, and how much time they have to save their lives and reduce damage to their assets. The approach proposed in this study can provide thisthese information.

Impact based forecasting has recently gained attention in disaster risk research (Merz et al., 2020;Taylor et al., 2018;Zhang et al., 2019). It aims at extending the forecast to include event impacts, such as the number and location of affected people and buildings, damage to buildings and infrastructure, or disruption of services. When obtaining specific and spatially resolved information on the expected event impact, as well as behavioural recommendations on what to do, people tend to be more motivated to accept warnings and to respond in a more effective way (Kreibich et al., 2021;Weyrich et al., 2018).

River flood forecasting systems, with lead times of several hours to days, are operational in many countries (e.g. Pappenberger et al., 2015). Flood warnings are commonly issued when given thresholds in terms of river water level or streamflow are exceeded. River flood impact forecasting systems have recently been proposed by Bachmann et al. (2016), Brown et al. (2016), and Dottori et al. (2017). One of the main challenges is the provision of timely and accurate estimates of inundation characteristics (Merz et al., 2020). To circumvent this simulation challenge, pre-defined relationships between river peak

discharges and expected impacts (Dale et al., 2016), and hydrodynamic surrogate model based on machine learning have been proposed (Hofmann and Schüttrumpf, 2020, 2021).

Germany, the Netherlands and Belgium have been hit by an extreme rainfall event in July 2021 leading to record-breaking peak flows at many gauges with estimated damage in the order of 30 billion € for Germany alone. Out of the 184 fatalities in Germany, 133 occurred along the river Ahr – a Rhine tributary[2]. Here, we show that a simple and rapid hydrodynamic flood

inundation model could extend the current hydrological forecasting systems by spatially explicit information on inundation areas, depths and flow velocities based on the forecasted gauge discharges or water levels. In that way, critical locations for life threatening flow conditions, for vehicle instabilities and structural failure of buildings and infrastructure could be derived from the inundation and flow velocity maps.

These maps could provide valuable and much more concrete information about the severity and the impact of the foreseen

flood event, which can be used for a more targeted disaster management. They can also assist in better warning and response recommendations for the population and thus help to reduce damages and particularly fatalities. We show that such an impact forecasting can be performed using a hydrodynamic model, that is easily setup based on readily available data, and has model runtimes that allow the application in operational flood forecasting and warning systems.

## 2 Hydrodynamic model

We implemented the hydrodynamic model *RIM2D* for the river Ahr river (Figure 1) in an hindcast setting for the July 2021 flood. *RIM2D* is a 2D raster-based model solving a simplified version of the shallow water equation, the so-called "local inertial approximation" of Bates et al. (2010). The approach has been used in a large number of applications of fluvial

---

[2] https://de.wikipedia.org/wiki/Hochwasser_in_West-_und_Mitteleuropa_2021

floodplain inundation and has been proven to provide realistic flow simulations (e.g. Falter et al., 2016;Neal et al., 2011). We selected this approach and implemented it in *RIM2D*, because  the 2D raster based concept using simplified shallow water

equations still provides the best compromise between required accuracy, model complexity and model runtime, as Bates (2022) states in a recent review on models for flood prediction.. As the original solution by Bates et al. (2010) is prone to instabilities for small grid cell sizes and under near-critical to super-critical flow conditions (de Almeida and Bates, 2013) often occurring during flash floods, the numerical diffusion as proposed by Almeida et al. (2012) was additionally implemented. This comes, however, at the cost of underestimated flow velocities (de Almeida and Bates, 2013). This effect becomes more pronounced

with increasing Froude numbers. For low dynamic fluvial floodplain inundation events with flow in the subcritical range these effects are negligible. Under flash flood conditions these limitations need to be considered when interpreting the inundation results. Simulated flow dynamics and velocities should be considered a low boundary estimation, with chances of higher velocities in reality (Shaw et al., 2020).

*RIM2D* implements the same numerical core as the well-known model Lisflood-FP (de Almeida and Bates, 2013), but is coded

in CUDA FORTRAN and implemented to run on large NVIDIA Tesla Graphical Processor Units (GPU). This enables massive parallelization of the numerical computations at low costs compared to large multi-core computing clusters.

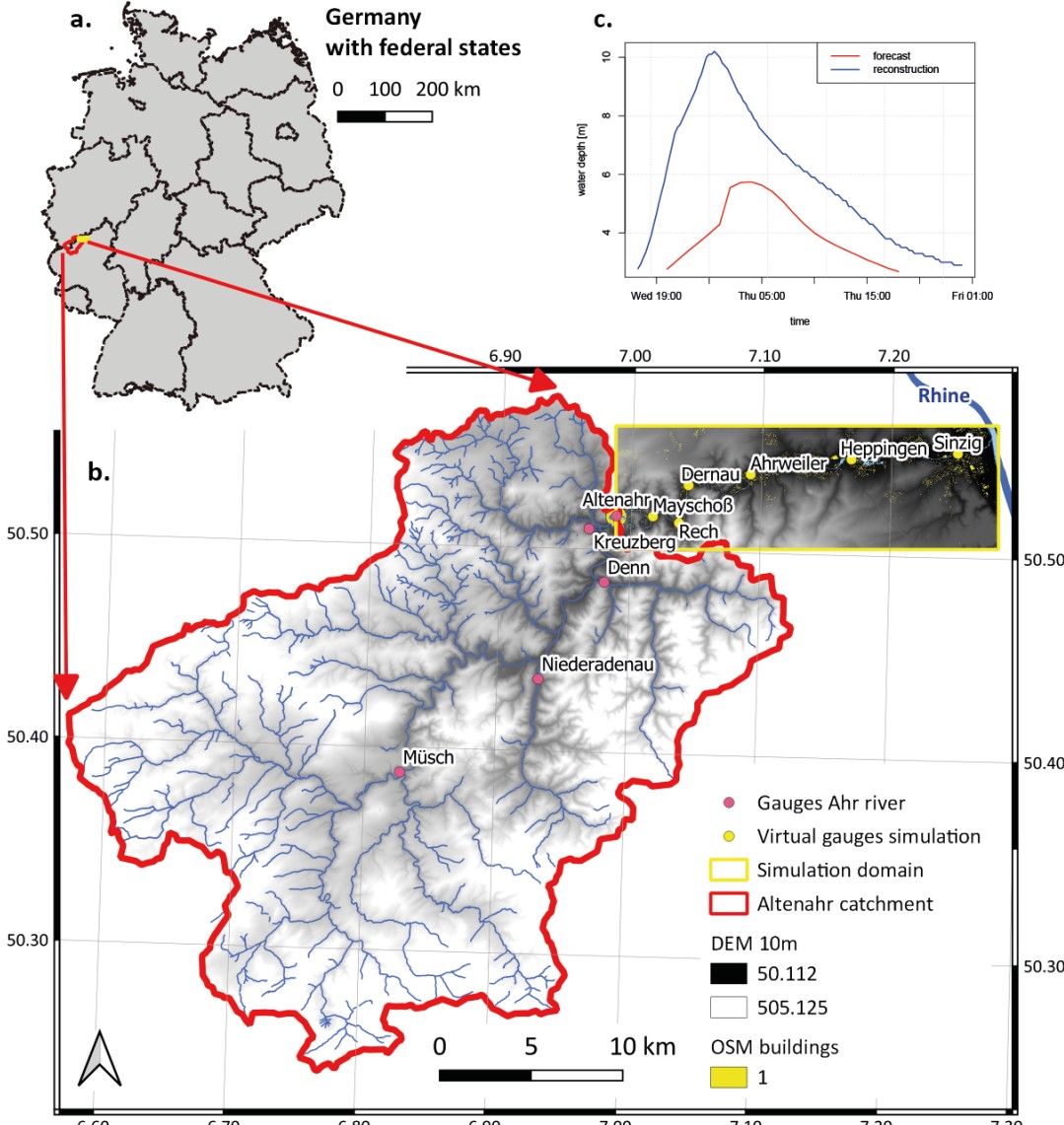

**Figure 1: Overview of the Ahr catchment and the simulation domain (river reach Altenahr – Sinzig). a. location of the catchment draining to the Altenahr gauge and the simulation domain within Germany. b. The catchment draining to the Altenahr gauge and the hydrodynamic simulation domain. c. Boundary water depths used for simulating the flood event 2021. "forecast" indicates the forecasted water depth hydrograph issued by the Landesumweltamt Rheinland-Pfalz, "reconstructed" indicates the preliminary (in January 2022) reconstructed water depth hydrograph at the gauge Altenahr of the event. The hydrographs start at July 14th and the time indicated at the time axis. Data sources: DEM 10 m resolution: © GeoBasis-DE / BKG [2012] (https://www.bkg.bund.de/); OSM river network and buildings: © OpenStreetMap contributors 2021. Distributed under the Open Data Commons Open Database License (ODbL) v1.0.**

## 3 Data requirements

*RIM2D* operates directly on spatial raster data in the format of ESRI ASCII raster. The core information is a Digital Elevation
Model (DEM) of a resolution suitable for the model domain and hydraulic situation to be simulated. For the selected model
domain of the reach of the river AR from the town Altenahr to Sinzig (i.e. to the inflow to the river Rhine, overall about 30
river kilometres) a resolution of 10 m was selected. This resulted in a raster grid with 675 rows and 2092 columns with overall
1,412,100 grid cells. The DEM was obtained from the German Federal Agency for Cartography and Geodesy (BKG). The
DEM was directly used as the basis for the flow simulation without any further modifications. This means that the river bed is
not realistically represented in the hydrodynamic model. The model river bed is rather a representation of the average water
surface of the river, which is in case of the Ahr typically less than one meter. This simplification is acceptable for the aim of
the model to simulate extreme flows exceeding the average flow depths by far. This approach is also justified by the fact, that
*RIM2D* operates with water levels, and not water depths or discharges as boundary conditions. Due to this, the water levels at
the model boundary will always be correct, despite the assumed bed elevation, and overbank flow and floodplain inundation
will be initiated at the right time and places. The benefit of this simplification is the applicability of the model approach to any
river reach without detailed local knowledge of river bathymetry. This enables an easy, semi-automated and low-cost
application and transfer of the model approach to practically everywhere, where a DEM with sufficient resolution in relation
to the river width is available.

*RIM2D* requires a hydraulic roughness parameterization. This was derived from the CORINE land use classification from the
European Environmental Agency, which is openly available for the whole of Europe. The raster data set was reclassified to 3
classes: build-up/sealed areas, forest and all other land use classes (farmland, pastures, water bodies, etc.). These classes were
assigned with typical Manning's roughness values from the literature: sealed surface areas: n = 0.02 (for simulating flow over
tarmac or concrete in the built-up areas), forest: n = 0.2, all other classes including the river channel and floodplains: n = 0.03.
The resulting spatially distributed roughness values were resampled to a raster with the same dimension and resolution as the
DEM. The validity of this classification approach has been shown in large-scale applications of Lisflood-FP (Bates, 2022;Wing
et al., 2021;Shaw et al., 2021;Bates et al., 2021;de Almeida et al., 2018;Savage et al., 2016;Stephens and Bates, 2015;de
Almeida and Bates, 2013;Almeida et al., 2012;Bates et al., 2010) and the hinterland inundation module of the Regional Flood
Model (RFM) (Vorogushyn et al., 2011;Falter et al., 2015;Sairam et al., 2021;Farrag et al., 2022), i.e. in studies where detailed
roughness calibration is not possible due to the size of the model domain and missing calibration data.

To simulate realistically the flow around buildings and in urban settings, the locations of buildings were extracted from the
Open Street Map (OSM) building layer. The vector shape file was rasterized to a grid with the same resolution and extent as
the DEM. The raster cells of the DEM where the building raster indicates a building are excluded from the hydraulic routing,
thus flow around buildings is simulated. This means that the model simulates the flow in the streets explicitly, and an
appropriate low roughness was selected for the built-up areas. This is in contrast to modelling approaches not considering the

buildings as obstacles, which thus requir a high roughness for the built-up area in order to compensate for the model simplification ("urban porosity approach") (Neelz and Pender, 2007).

Initial water depths were derived by a steady-state simulation prescribing a fixed water level at the inflow of the river channel into the modeling domain, with free outflow, i.e. normal depth, at the lower boundary. The simulation was continued until a constant water profile along the river reach was established. As a consequence of this procedure and the missing bathymetry,

only water levels exceeding the water levels of the initial water depths can be simulated. This is, however, acceptable for the purpose of simulating flood flows largely exceeding the average river flow. In order to test the sensitivity of the inundation simulation to the missing river bed, a control simulation was performed, in which the DEM elevation of the initially wet cells, i.e. the river bed in the model, is lowered by the water depths corresponding to the 2-year return period flow at the gauges in the reach. For both gauges this amounts to 0.85 m, thus the elevation of the wet cells was reduced uniformly by 0.85 m. This

depth corresponding to the bankfull discharge is a conservative estimate for the channel depth below the water surface not included in the DEM.

As input to *RIM2D* we used the official water level forecast of the flood warning centre Rhineland-Palatinate at the gauge Altenahr and the reconstruction of the actual water levels in meter above sea level. The reconstruction was necessary because the gauge was destroyed during the event. For the actual fluvial flood simulations these water levels are prescribed to the

inflow cells into the domain. These cells were set on the river channel on the boundary of the domain, which is clearly visible in the DEM. In order to account for overbank flow also those cells neighboring the river channel and with elevations below the maximum water level of the flood hydrograph were additionally selected. Water depths were assigned to those cells only when the river water levels exceeded the DEM elevation. The forecast of the water levels at gauge Altenahr was issued with a lead time of 24 hours before the flood event, with a maximum water depth of 5.74 m and a hydrograph duration of 30 hours.

In order to validate the simulation results, an additional simulation using the preliminary hydrograph (in January 2022) of the flood event at gauge Altenahr reconstructed by the Landesumweltamt Rheinland-Pfalz (LfU)[3] was performed. This shows a peak water depth of 10.2 m (**Figure 1c**), i.e. 4.46 m higher than the forecast. The large difference between the forecast and the reconstruction is not only caused by an underestimation of the flow by the hydrological forecast, but to a large extent also by clogging of bridges, one of which is directly located downstream of the gauge Altenahr.

## 4 Results and discussion


*Figure 2* shows the simulated maximum water depths for the flood forecast hydrograph and the reconstructed hydrograph as inflow boundary. Both maps show large inundation areas, particularly in the towns and villages situated in the floodplains alongside the river Ahr. The inundation depths and extent of the reconstructed scenario are, however, much higher than those

---

[3] Environmental Office of the federal state Rhineland-Palatinate

from the forecast, because of the higher peak water level. This is illustrated by the inundation depths for the heavily affected commune of Ahrweiler (inset of *Figure 2b*).

The model performance was validated using the post-event mapping of the inundated areas by the LfU, which were compared to the simulated inundation area based on the reconstructed hydrograph (*Figure 2b*). A high agreement of the simulation with the maps can be observed, supporting confidence in the simulation results. The binary pattern comparison metric $F^{(2)}$ as proposed by Aronica et al. (2002), also termed Critical Success Index or Threat Score (Horritt et al., 2007;Sampson et al., 2015;Lim and Brandt, 2019), evaluates to 0.845, which is a very high performance value for hydrodynamic inundation simulations. Furthermore, water depths derived from 75 high water marks at buildings reported by the inhabitants (red dots in *Figure 2b*) were used for the evaluation of the simulated water depths. In this context it is noteworthy that the reported water depths refer to different vertical datums, like the street, the pedestrian walk or the doorstep, which needs to be considered in the evaluation of the comparison. The bias between the reported water depths and the simulated evaluated to -0.39 m, with an RMSE of 0.66 m. Considering the uncertainty in the datum of the reported water depths, the unavoidable simplification of the terrain in the used 10 m resolution DEM, and the uncalibrated simulation such differences can be excepted. Considering the inundation water depths between 1.5 and 3 m in the area where water marks were recorded, the model performance is considered good and sufficient for an early warning. Almost identical results are obtained with the simulations on a lowered rived bed (Supplement S1). The lowered bed has a noticeable impact on channel water depths and flow velocities, but little on the overbank flow und floodplain inundation. The channel capacity is hence comparably small in relation to flood volume. The presented model performance thus supports the validity of the simplified assumptions in the model setup, including the non-consideration of the real river bed bathymetry.

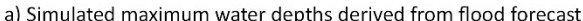

a) Simulated maximum water depths derived from flood forecast

b) Simulated maximum water depths derived from reconstructed water levels Altenahr

**Flood July 2021 simulated max. inundation depth [m]**

- 0.10
- 0.30
- 0.60
- 1.00
- 1.50
- 3.00
- 4.00
- 12.00
- mapped inundatiom

**OSM**
- building
- roads

**DEM 10m**
- 22
- 400
- water marks

**Figure 2. Maximum inundation depths during the flood in July 2021 simulated with RIM2D using a) the water level forecast for the gauge Altenahr, and b) the reconstructed hydrograph of the event as shown in Figure 1. The green outlined areas indicate the inundation areas mapped by the LfU of Rhineland-Palatinate. Data sources: DEM 10 m resolution: © GeoBasis-DE / BKG [2012] (https://www.bkg.bund.de/); OSM roads and buildings: © OpenStreetMap contributors 2021. Distributed under the Open Data Commons Open Database License (ODbL) v1.0.**

*Figure 3a* shows the maximum simulated effective flow velocities using the reconstructed flood hydrograph. The effective flow velocities were quantified as the geometric vector sum of the flow velocities in the x- and y-directions of the grid. The simulated flow velocities are plausible for such a dynamic event, ranging from 2 m/s to 5 m/s in the river course, and 0.1 m/s to 2 m/s in the build-up areas. The model typically simulates increased flow velocities between the buildings, which is plausible for flow in an urban setting (inset in *Figure 3a*).

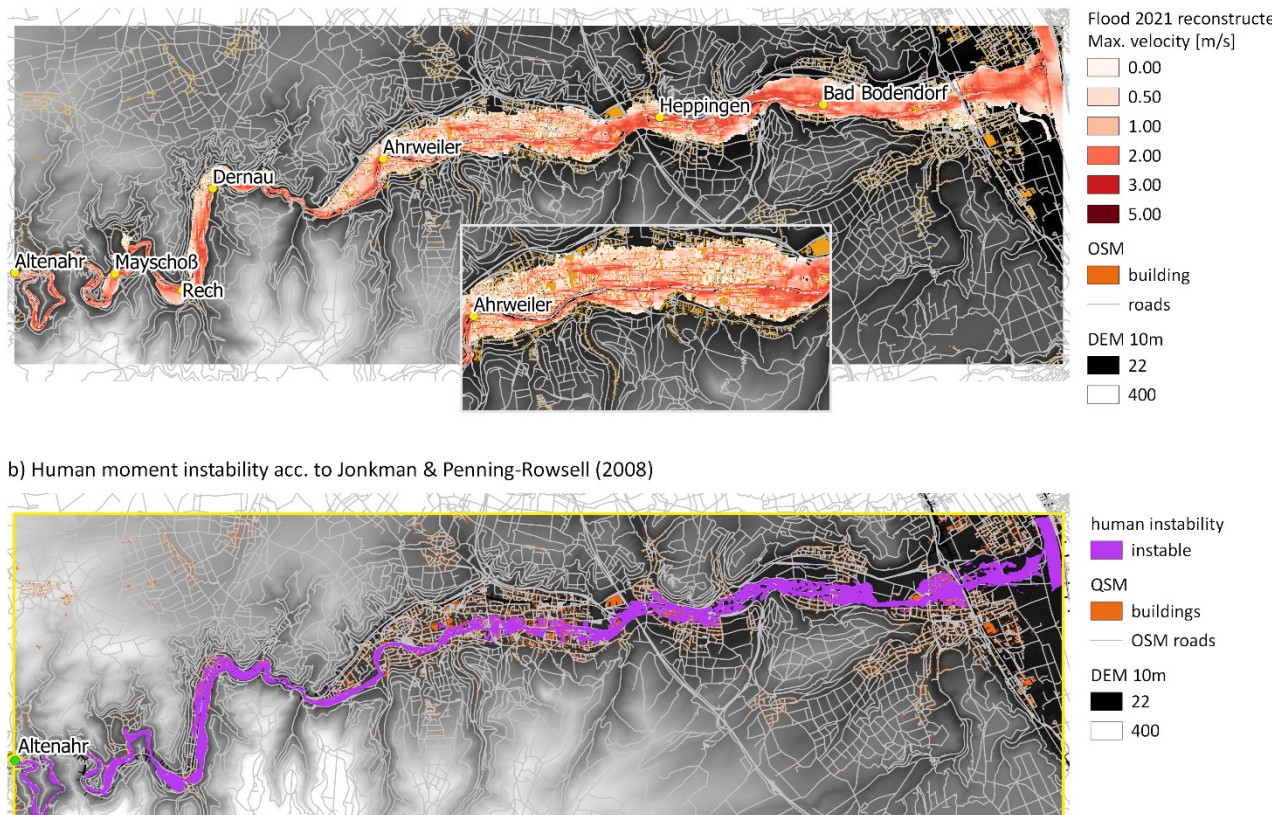

a) Simulated maximum flow velocities based on the reconstructed hydrograph Altenahr

b) Human moment instability acc. to Jonkman & Penning-Rowsell (2008)

**Figure 3. Flow velocities and human instability indicator for the flood event in July 2021 using the reconstructed hydrograph: a) simulated maximum effective flow velocities; b) areas indicating human moment instability in flowing water derived from the maximum values of the product of water depth and flow velocity. The critical value for human instability was set to 1 m²/s, following Jonkman and Penning-Rowsell (2008) and considering potential underestimation of flow velocities by RIM2D. Data sources: DEM 10 m resolution: © GeoBasis-DE / BKG [2012] (https://www.bkg.bund.de/); OSM river network and buildings: © OpenStreetMap contributors 2021. Distributed under the Open Data Commons Open Database License (ODbL) v1.0.**

As an example for an additional impact forecast based on the hydraulic simulations, a location-specific indicator for human instability was derived as a product of simulated inundation depths and flow velocities. Jonkman and Penning-Rowsell (2008) reported the critical threshold of moment instability for humans in water flows at 1.32 m²/s. Hence, even more detailed warnings could be issued for humans losing control in flowing water and being carried away with a high risk of drowning. An example is shown in *Figure 3b*, in which the maximum value of the product of water depth and flow velocity is used as indicator for human instability. Here, a conservative critical value of 1 m²/s was chosen to account for possible underestimation of the flow velocities caused by the numerical approach of the model (cf. section 2), but also for differences in person weights

and fitness, and to consider variation in the roughness of the ground. Alternative approaches for estimating the risk of human fatalities in floods are published by REDSCAM (2000), Penning-Rowsell et al. (2005), Jonkman et al. (2008), or Milanesi et al. (2015). Additional indicator maps can be derived from the water depth and flow simulations, e.g. for vehicle instability using the approaches of Bocanegra and Francés (2021), Martínez-Gomariz et al. (2019), or Milanesi and Pilotti (2020), or for

structural failure of buildings using the approach of Kelman and Spence (2004) or Jansen et al. (2020). These maps can be automatically derived from the simulation results, i.e. could be made available along with the forecast and inundation maps. It is also possible to use the maps of water depths and velocity as input into impact models that combine exposure and vulnerability information. In this way, direct or indirect adverse consequences, such as failure of critical infrastructure or economic loss to buildings and infrastructure, could be estimated (Merz et al., 2020;Rözer et al., 2021).

*Computational performance*

The 30-hour long flood event was simulated in 14 minutes on a NVIDIA TESLA P100 GPU computing unit connected to a Linux server with Intel Xeon Gold 6140 CPU. This is a simulation runtime equivalent to less than 0.8% of the simulated event duration. The memory capacity of the GPU unit in terms of computational nodes was used to about 15% only, leaving room for increasing the model domain or spatial resolution. The achieved simulation runtimes would allow using the model in an

operational flood forecast mode. With a lead time of 24 hours the simulated inundation areas could be available more than 23 hours prior to the event. This would leave sufficient time to include the inundation, flow velocity and indicator maps in the emergency management, and to provide informative and localized warnings to the population.

**Uncertainties**

As with any model simulation, there are uncertainties associated to the results. The presented uncalibrated hydrodynamic model will surely add to the already existing uncertainties of the meteorological and hydrological forecasts. However, inferring from previous studies about the uncertainties in flood risk assessments (Apel et al., 2009;de Moel and Aerts, 2011), where the uncertainty added by the hydrodynamic modeling was identified as the smallest among different uncertainty sources, the forecast uncertainty added by the hydrodynamic modeling can be assumed small compared to the uncertainty that is contained

in the water level forecast. This uncertainty can be further reduced, if the hydrodynamic model is not setup ad-hoc and run uncalibrated as in this feasibility study, but setup considering the bathymetry in more detail and also calibrated or validated against historic floods events. Such a more detailed implementation could be easily achieved with the local knowledge of the responsible authorities of the river reaches.

The evaluation of the forecast skill, e.g. by a ROC curve and score would also be desirable, but considering the rarity of such

extreme and documented flood events at a particular river reach, it is practically not possible to evaluate forecast skill of the whole forecast chain (meteorology – hydrology – hydraulics), simply because of lack of event data.

For the use in an operational setting, it is also advisable to provide uncertainty maps derived from an ensemble of hydrological forecasts to map the uncertainties inherent in the meteorological forecasts and the hydrological modelling. Because of the computational efficiency of *RIM2D* these hydrological forecast ensembles can be transferred into probabilistic inundation

maps mapping the consequences of the uncertainties in the meteorological and hydrological forecasts as uncertainty in the inundation forecast. However, currently the hydrological forecasts for river gauges in Germany do not provide this kind of information, but only the most likely flood hydrograph, which was used for the flood simulation shown in Figure 2a.

## 5 Conclusions

The recent flood disaster in West Germany in July 2021 is used to demonstrate the potential benefits of flood impact
forecasting. We show that the simplified and easily setup hydrodynamic model *RIM2D*, that uses readily available data, delivers plausible inundation areas, depth and flow velocity simulations in runtimes enabling forecasts and early warning. Moreover, additional impact indicators identifying dangerous hotspots, for instance, in terms of expected building collapse, persons drowning, or floating and toppling cars can be derived. Such detailed and location-specific information on expected impacts is highly valuable for a targeted and spatially explicit flood disaster management. It also allows more meaningful
warnings of the population compared to the standard, gauge-based water level forecast. We argue that the use of this information can substantially improve the current disaster management and warning response. People lives can be saved, even if the hydrological water level forecasts underestimate the actual event, as was the case in the July 2021 event (cf. Figure 2a). We believe that the disaster management could have been more targeted as it actually was, if the information provided by the simulation based on the hydrological forecast as shown in Figure 2a would have been available prior to the event. Moreover,
it can be hypothesized that the early warning would have made a deeper impact in the affected population and the disaster management units, thus likely reducing the extraordinary high number of fatalities, if communicated in due time.

Due to the model implementation of *RIM2D* on Graphical Processor Units dedicated for massive parallel computing, the simulation runtimes are in a range suitable for inclusion in an operational flood early warning system. As the model setup is simple and the required data are readily available in many countries, the model can be widely transferred and used. The required
hardware environment for the simulation is affordable at low costs, particularly in comparison with large-scale computational clusters. This facilitates implementation of the model at flood forecast centres without considerable investments into large-scale IT infrastructure. Based on the validity of the simulations, the ease of implementing hydraulic forecast models and the speed of the simulations, we argue that the current forecast practices should be extended with impact forecast models as presented here in order to improve the efficiency of flood warnings and management. The scientific basis, methods and models
for these impact forecasts are mature for implementation in operational forecast systems.

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
