# Peer review of "Brief communication - Impact Forecasting Could Substantially Improve the Emergency Management of Deadly Floods: Case Study July 2021 floods in Germany"

_Natural Hazards and Earth System Sciences, 2022_

## Author Response (AR1)

**Author responses to referee comments on "Brief communication – Impact Forecasting Could Substantially Improve the Emergency Management of Deadly Floods: Case Study July 2021 floods in Germany" by Heiko Apel et al., Nat. Hazards Earth Syst. Sci. Discuss., https://doi.org/10.5194/nhess-2022-33-RC1, 2022**

In the following the original reviewer comments are marked in blue.

**Author response to the comment of the anonymous referee #1**

We thank the referee to take the time to review our manuscript, and the general judgement that it is a valuable contribution. However, we cannot follow the critical comments, which are mainly based on the wrong assumption that the model is not validated.

Referee comment: You show results of your model run, but not a validation. Do you have the capability to conduct a validation of any sort for the RIM2D simulations? This validation can be from a historic event; it does not have to be for the event discussed in the paper. Nonetheless, some validation is required in order to show that the model is producing realistic results.

As we describe on page 5 in lines 122-132 and shown in figure 2 the model is validated against the actual flood event of 2021. We compared the model results to the latest and most valid mapping of the flood extent and calculated the binary flood pattern metric $F^{(2)}$, which is a standard for the quantification of hydraulic model performance ($F^{(2)}$ is also known as Critical Success Index CIS or threat score (Horritt et al., 2007;Sampson et al., 2015;Lim and Brandt, 2019)). This evaluated to a high value of 0.845. Moreover, we validated the simulated inundation depths against 75 surveyed high flood water marks, which resulted in a minimum bias of 0.09 m, and a RMSE of 0.30 m. Thus, the model setup for this river reach, including the assumption taken for the river bed, is well validated. In fact, it is much better validated than many other hydraulic simulations, for which the validation data as presented here is missing.

Referee comment: Since you don't have riverbed data, you assume that the mean river surface is the bed. You state that this is an acceptable assumption because the flooding river is much deeper than the river under normal conditions. Please conduct a sensitivity analysis to quantify this: under what situations can this approximation be made, and what is the error it introduces? You can run your simulation with a range of river bed depths assumed to quantify this as a sensitivity analysis.

We see the good model performance in both simulating the inundation extend as well as inundation depths as an empirical proof for our statement. Moreover, the assumption can also be theoretically justified by the model setup. Reducing the river bed as mapped in the DEM, in this case in the range of -0.1 m to max -0.8 m, would not change the simulation results, particularly not the simulated inundation extent. This can be deduced from the model setup: RIM2D operates with water levels for fluvial boundary conditions. Thus, the water levels at the model boundary will always be correct, despite the assumed bed elevation. This would be different when discharge is used as boundary condition. These realistic water levels at the boundary ensure realistic flood propagation simulation, because overbank flow is simulated at the right time and right locations. And this is of major importance for realistic overbank inundation simulations. The actual discharge in the river bed is of less importance for this, and once the flow is overbank, the out-of-bed topography determines the spatial inundation, not the bed topography. Again, it has to be noted here that the proposed approach

targets floods largely exceeding in-bed flow, not detailed low flow. We therefor see no gain in performing a sensitivity analysis varying river bed elevation. Moreover, including this in the manuscript would shift the focus away from the actual message of the paper, that is the large gain in flood forecast information by using hydraulic model flood forecasts.

Referee comment: It would also be good to carry out a sensitivity analysis to Mannings n values chosen. This is because the n=0.05 use for forests seems a bit small. Arcement & Schneider (1984) and Petryk & Bosmajian (1975) show n values between 0.1 and 0.2 for flooded forests.

We regard the suggested sensitivity analysis of river bed elevations and roughness as an exercise, that does not create additional knowledge. The mathematical and numeric foundation of the RIM2D model has been shown in many applications and papers using LISFLOOD-FP, to which RIM2D is identical in terms of numerical scheme and hydraulic equations. To cite just a few papers, we refer to (Bates, 2022;Wing et al., 2021;Shaw et al., 2021;Bates et al., 2021;de Almeida et al., 2018;Savage et al., 2016;Stephens and Bates, 2015;de Almeida and Bates, 2013;Almeida et al., 2012;Bates et al., 2010). Thus we see no need and particularly gain in performing sensitivity analysis.

Referee comment: You use a threshold value for human instability. There are many values published in the literature, also as a function of many different parameters (depth, depth-speed product, depth-speed^2 product). You should do a sensitivity analysis to this. In addition to human instability, why not also look at vehicle instability? A good summary of the results in this field is given by Martinez-Gomariz et al. (2016). Another useful thing to assess would be building collapse. For example Jansen et al. (2020). As with human instability, the literature includes a wide range of results, so a sensitivity analysis would be needed, but it would strengthen the utility of this paper.

The maps showing the flood impacts are meant to show the potential of an additional value of spatially explicit flood forecast by a hydrodynamic model. We chose the approach of Jonkman and Penning-Rowsell (2008) to illustrate this potential gain exemplarily. Of course, other approaches can be used, which might result in slightly different maps. But the main message is here that with the proposed approach, this kind of maps can be provided rapidly to flood emergency managers. This is the decisive message here and much more important than potential differences in the estimation of human instabilities caused by different methods. And of course, other impact indicators like the mentioned car instability, or potential structural damage to houses can additionally be provided, and have actually been derived from the simulations. We have done this for cars using the approach of Bocanegra and Francés (2021), but it is not shown because of the limitation to 3 figures in the brief communications format. But again, the main message is here, that these impact forecasts are only possible using the hydraulic model for flood forecasting. Without this approach no spatial explicit impact forecasting would be possible at all. The shown impact on human instability is used to illustrate this, and is not meant as the only possible impact to be forecasted. We will stress this point in more detail in a revised version of the manuscript.

References

Almeida, G. A. M. d., Bates, P., Freer, J. E., and Souvignet, M.: Improving the stability of a simple formulation of the shallow water equations for 2-D flood modeling, Water Resources Research, 48, W05528, doi:10.1029/2011WR011570, 2012.

Bates, P. D., Horritt, M. S., and Fewtrell, T. J.: A simple inertial formulation of the shallow water equations for efficient two-dimensional flood inundation modelling, Journal of Hydrology, 387, 33-45, DOI 10.1016/j.jhydrol.2010.03.027, 2010.

Bates, P. D., Quinn, N., Sampson, C., Smith, A., Wing, O., Sosa, J., Savage, J., Olcese, G., Neal, J., Schumann, G., Giustarini, L., Coxon, G., Porter, J. R., Amodeo, M. F., Chu, Z., Lewis-Gruss, S., Freeman, N. B., Houser, T., Delgado, M., Hamidi, A., Bolliger, I., E. McCusker, K., Emanuel, K., Ferreira, C. M., Khalid, A., Haigh, I. D., Couasnon, A., E. Kopp, R., Hsiang, S., and Krajewski, W. F.: Combined Modeling of US Fluvial, Pluvial, and Coastal Flood Hazard Under Current and Future Climates, Water Resources Research, 57, e2020WR028673, https://doi.org/10.1029/2020WR028673, 2021.

Bates, P. D.: Flood Inundation Prediction, Annual Review of Fluid Mechanics, 54, 287-315, 10.1146/annurev-fluid-030121-113138, 2022.

Bocanegra, R. A., and Francés, F.: Assessing the risk of vehicle instability due to flooding, Journal of Flood Risk Management, 14, e12738, https://doi.org/10.1111/jfr3.12738, 2021.

de Almeida, G. A. M., and Bates, P.: Applicability of the local inertial approximation of the shallow water equations to flood modeling, Water Resources Research, 49, 4833-4844, 10.1002/wrcr.20366, 2013.

de Almeida, G. A. M., Bates, P., and Ozdemir, H.: Modelling urban floods at submetre resolution: challenges or opportunities for flood risk management?, Journal of Flood Risk Management, 11, S855-S865, 10.1111/jfr3.12276, 2018.

Jonkman, S. N., and Penning-Rowsell, E.: Human Instability in Flood Flows, JAWRA Journal of the American Water Resources Association, 44, 1208-1218, https://doi.org/10.1111/j.1752-1688.2008.00217.x, 2008.

Savage, J. T. S., Bates, P., Freer, J., Neal, J., and Aronica, G.: When does spatial resolution become spurious in probabilistic flood inundation predictions?, Hydrological Processes, 30, 2014-2032, 10.1002/hyp.10749, 2016.

Shaw, J., Kesserwani, G., Neal, J., Bates, P., and Sharifian, M. K.: LISFLOOD-FP 8.0: the new discontinuous Galerkin shallow-water solver for multi-core CPUs and GPUs, Geosci. Model Dev., 14, 3577-3602, 10.5194/gmd-14-3577-2021, 2021.

Stephens, E., and Bates, P.: Assessing the reliability of probabilistic flood inundation model predictions of the 2009 Cockermouth, UK, Hydrological Processes, n/a-n/a, 10.1002/hyp.10451, 2015.

Wing, O. E. J., Smith, A. M., Marston, M. L., Porter, J. R., Amodeo, M. F., Sampson, C. C., and Bates, P. D.: Simulating historical flood events at the continental scale: observational validation of a large-scale hydrodynamic model, Nat. Hazards Earth Syst. Sci., 21, 559-575, 10.5194/nhess-21-559-2021, 2021.

**Author response to comments of referee #2**

We thank the reviewer for his/her thoughtful comments on the manuscript. The different comments are addressed individually below.

Referee comment: The first is the claim on Line 20-22 that river flood forecasts are typically provided only at river gauges. Since I rarely work with forecasting I may be wrong, but as far as I am aware there are existing European flood forecasts that already provide spatially distributed data, from the global Copernicus Emergency Management Service to specialized tools for riverine flooding. So adding a few lines and references that justifies the statement would be an advantage.

Our statement refers to the official and legally binding operational flood forecasts in Germany, operated be the different federal states. All these flood forecasts provide only forecasts of water levels or discharges at a selection of river gauges, derived from weather forecasts and hydrological models. The same holds true for Global Flood Awareness System GloFAS, developed by the European Commission and the European Centre for Medium-Range Weather Forecasts (ECMWF). On a European

level the European Flood Alert System EFAS (https://www.efas.eu) provides warnings with spatial information. However, these forecasts including inundation areas have a rather coarse spatial resolution of 100 m, do not consider dikes or other flood protection measures, and are based on pre-calculated hazard maps, not actual flood dynamics. These forecasts are thus valuable for generating a general flood alert and indicating the affected areas in the expected flood, but for detailed flood management actions within the flood areas they are of limited use. For targeted flood management actions and disaster response much more detailed information as provided by the hydraulic model in this manuscript are required.

We will add an explanation of the novelty of our work compared to existing flood warning systems in the revised manuscript, to underline the necessity of more detailed inundation and impact simulations for flood management.

Referee comment: More importantly the choice of simplified 2D model seems rather arbitrary (L50). Much has happened over the past 12 years and at least 5 European research institutions have worked on the field suggesting a wide number of models and also outside of Europe this is an research field. So justifying the choice of tool should go beyond a subjective assessment of the model performance using a single metric stating that the model is sufficiently accurate. As far as I know the most recent review of methods is in Thrysøe et al (2021).

It is correct that research has provided a number of potential methods and models for high resolution inundation simulations in the past decade. The presented model RIM2D is one of these models. However, the choice of this particular modelling approach, which is from the underlying mathematical concept identical to Lisflood-FP, is not arbitrary. As already shown in Apel et al. (2009) and many studies later, the 2D raster based concept using simplified shallow water equations still provide the best compromise between required accuracy, model complexity and model runtime. Bates (2022) evaluated many studies on flood prediction models in the latest review and concluded: "…there is a broad consensus, among researchers at least, that 2D models are the current best compromise between what we know about the physics of inundation, the compute resources we have, and the data currently available.". The RIM2D model selected for this study falls exactly in this category, being computationally efficient, being parsimonious by requiring only a limited and easily accessible data for model setup and running, and by short simulation times due to its massive parallelization on GPUs. We will add a justification for the selection of RIM2D in the revised manuscript, based on the arguments listed above.

But we would surely acknowledge that other, similarly efficient and accurate flood simulation models can be used. The core message of this contribution is, that by providing spatially distributed and high-resolution inundation maps and derived flood impact maps in a flood forecast would improve the effectiveness of flood warnings, and eventually save lives. We will underline this in the revised manuscript.

Referee comment: My last point is that it is not clear from the paper how the input data are related to the forecasts. Precipitation forecasts are inherently quite uncertain as the authors correctly state (L174-175). Nevertheless the study seems to use a reconstruction of data that ignores this important source of uncertainty. A forecast should hence include this uncertainty and then there is an additional discussion of e.g. ROC curves. Hence the title of the paper should be adjusted to reflect the fact that the paper mainly focus on providing a spatial distribution of a given flow than to provide an impact forecast where the uncertainty of the forecast is also included.

The reconstructed hydrograph was used for model validation. By using the currently best estimate of the actual flood event, the high validity of RIM2D and the simulation results was shown. This would not be the input to the model in a forecast mode. In a forecast mode the output of the hydrological model, that transfers the rainfall forecasts into hydrographs, would be used to drive RIM2D. An example of the forecast mode is shown in figure 2a, where the actual operational forecast was used as boundary condition for the hydraulic model. The comparison with simulation based on the reconstructed hydrograph in figure 2b provides an estimation of the uncertainty of the river discharge forecast stemming from the meteorological forecast and hydrological modelling, expressed as an underestimation of the actual inundation extent and depths. We agree that ideally a spatial flood forecast should provide an uncertainty band. This can be achieved, if not only the ensemble mean of the river discharge forecasts is used, but the whole or a selection of the forecast ensemble. The uncertainty can then be expressed as probability of inundation for each grid cell and statistics of inundation depths and flow velocities per grid cell. This is generally feasible, but requires larger computational resources in terms of multiple GPUs in order to run the ensemble forecasts in parallel.

A ROC curve and score would also be desirable, but considering the rarity of such large flood events, it is practically not possible to evaluate the ROC or similar forecast quality measures, simply because of lack of event data.

For the presented manuscript, no ensemble meteorological and hydrological forecast were available. This impedes the assessment of uncertainty through ensemble integration. But nevertheless, we would still argue that a spatially explicit flood forecast based on the ensemble mean, i.e. without uncertainty information, would be a tremendous step forward towards a more informed and targeted disaster management compared to the current state. A flood warning as shown in Figure 2a, even if it underestimated the actual flood extent, would have surely raised more concerns and alertness in the disaster management centres, and provide the required information for targeted actions, as e.g. evacuations, flood protection measures, affected critical infrastructure, etc. And this is the main message of this contribution.

In the revised manuscript we will address the point of uncertainty and ensembles in the discussion, following the arguments above.

Almeida, G. A. M. d., Bates, P., Freer, J. E., and Souvignet, M.: Improving the stability of a simple formulation of the shallow water equations for 2-D flood modeling, Water Resources Research, 48, W05528, doi:10.1029/2011WR011570, 2012.

Apel, H., Aronica, G., Kreibich, H., and Thieken, A.: Flood risk analyses—how detailed do we need to be?, Natural Hazards, 49, 79-98, 2009.

Bates, P. D., Horritt, M. S., and Fewtrell, T. J.: A simple inertial formulation of the shallow water equations for efficient two-dimensional flood inundation modelling, Journal of Hydrology, 387, 33-45, DOI 10.1016/j.jhydrol.2010.03.027, 2010.

Bates, P. D., Quinn, N., Sampson, C., Smith, A., Wing, O., Sosa, J., Savage, J., Olcese, G., Neal, J., Schumann, G., Giustarini, L., Coxon, G., Porter, J. R., Amodeo, M. F., Chu, Z., Lewis-Gruss, S., Freeman, N. B., Houser, T., Delgado, M., Hamidi, A., Bolliger, I., E. McCusker, K., Emanuel, K., Ferreira, C. M., Khalid, A., Haigh, I. D., Couasnon, A., E. Kopp, R., Hsiang, S., and Krajewski, W. F.: Combined Modeling of US Fluvial, Pluvial, and Coastal Flood Hazard Under Current and Future Climates, Water Resources Research, 57, e2020WR028673, https://doi.org/10.1029/2020WR028673, 2021.

Bates, P. D.: Flood Inundation Prediction, Annual Review of Fluid Mechanics, 54, 287-315, 10.1146/annurev-fluid-030121-113138, 2022.

Bocanegra, R. A., and Francés, F.: Assessing the risk of vehicle instability due to flooding, Journal of Flood Risk Management, 14, e12738, https://doi.org/10.1111/jfr3.12738, 2021.

de Almeida, G. A. M., and Bates, P.: Applicability of the local inertial approximation of the shallow water equations to flood modeling, Water Resources Research, 49, 4833-4844, 10.1002/wrcr.20366, 2013.

de Almeida, G. A. M., Bates, P., and Ozdemir, H.: Modelling urban floods at submetre resolution: challenges or opportunities for flood risk management?, Journal of Flood Risk Management, 11, S855-S865, 10.1111/jfr3.12276, 2018.

Horritt, M. S., Di Baldassare, G., Bates, P. D., and Brath, A.: Comparing the performance of a 2-D finite element and a 2-D fintite volume model of floodplain inundation using airborne SAR imagery, Hydrological Processes, 21, 2745-2759, 2007.

Jonkman, S. N., and Penning-Rowsell, E.: Human Instability in Flood Flows, JAWRA Journal of the American Water Resources Association, 44, 1208-1218, https://doi.org/10.1111/j.1752-1688.2008.00217.x, 2008.

Lim, N. J., and Brandt, S. A.: Flood map boundary sensitivity due to combined effects of DEM resolution and roughness in relation to model performance, Geomatics, Natural Hazards and Risk, 10, 1613-1647, 10.1080/19475705.2019.1604573, 2019.

Sampson, C. C., Smith, A. M., Bates, P. B., Neal, J. C., Alfieri, L., and Freer, J. E.: A high-resolution global flood hazard model, Water Resources Research, 51, 7358–7381, 10.1002/2015WR016954, 2015.

Savage, J. T. S., Bates, P., Freer, J., Neal, J., and Aronica, G.: When does spatial resolution become spurious in probabilistic flood inundation predictions?, Hydrological Processes, 30, 2014-2032, 10.1002/hyp.10749, 2016.

Shaw, J., Kesserwani, G., Neal, J., Bates, P., and Sharifian, M. K.: LISFLOOD-FP 8.0: the new discontinuous Galerkin shallow-water solver for multi-core CPUs and GPUs, Geosci. Model Dev., 14, 3577-3602, 10.5194/gmd-14-3577-2021, 2021.

Stephens, E., and Bates, P.: Assessing the reliability of probabilistic flood inundation model predictions of the 2009 Cockermouth, UK, Hydrological Processes, n/a-n/a, 10.1002/hyp.10451, 2015.

Wing, O. E. J., Smith, A. M., Marston, M. L., Porter, J. R., Amodeo, M. F., Sampson, C. C., and Bates, P. D.: Simulating historical flood events at the continental scale: observational validation of a large-scale hydrodynamic model, Nat. Hazards Earth Syst. Sci., 21, 559-575, 10.5194/nhess-21-559-2021, 2021.

---

## Author Response (AR2)

**Response to review, second revision round – review 1**

The review comments are marked in blue, our response in black.

With regard to river bed elevation, you state in your response that this will not have an effect on your result, as you are not investigating low flows. Can you state a range of depths that the channel is expected to vary? That is, are the non-flood channels only a few cm deep, so that the flood depth is much greater than the non-flood channel depth? Or are they similar order of magnitude? I'd expect that channel depth has a large effect on the transit time of the flood wave, as well as the diffusion of the flood wave. Therefore, I do not agree with the authors' reply that channel depth is irrelevant, and I still think a sensitivity analysis should be carried out. The authors' other response that depth is irrelevant because the downstream boundary condition is normal depth, is only applicable for steady flow conditions, not unsteady flow, so I don't think that answers the question.

Under normal flow conditions the channel depths are in the range of 50 cm at the two gauges in the studied river reach. The estimated flow depth of the 2-year event is 85 cm at both gauges (Altenahr and Bad Bodendorf). In this upper part of the reach with a deeply incised river into steep hills, the depth of 85 cm is an order of magnitude lower than the flow depths observed during the flood event, which reached depths of 8 – 12 m. In the lower part of the reach the flow depths are lower, but still in the range of 4 – 6 m, thus still much larger than the flow depth of the 2-year flood, almost an order of magnitude.  This is the basis of our argument, that the neglected river bed has little effect on the simulation of the overall flood extent and floodplain inundation depths.

In order to show the effect of the neglected river bed, we corrected the DEM cells identified as river bed. We hereby conservatively assume that the LiDAR DEM refers to flow depths of the 2-year flood, i.e. bankfull discharge. Thus, we lowered the DEM of the river bed by 85 cm and repeated the simulation to show the effect of the neglected river bed. The obtained results differ only marginally from the results obtained with the original model version. Figure 1 shows the difference in maximum inundation depths, and Figure 2 shows the difference in maximum flow velocities. It can be seen that the differences in water depths are mainly in the range of -0.1 to 0.1 m, and the differences in flow velocities are in the same range in m/s. This is also illustrated by the bottom right histogram in Figure 3 showing the cell-wise differences of the maximum inundation depths. Larger differences occur, as expected, in the identified river bed of the DEM. However, the floodplain inundation depths and flow velocities, as well as the inundation extent are practically identical. This is also expressed in the almost identical model performances listed in Table 1, comparing the observed inundation extent with the simulated by the Critical Success Index CSI, and the performance in simulating inundation depths by comparing the 75 reported flood marks with the maximum inundation depths (Bias and RMSE). It has to be noted that the performance values slightly modified compared to the manuscript, due to an error in projection of the shape file showing the locations of the water marks, which was used for extracting the water depths at the locations of the water masks from the simulation results.

Overall, the simulation of the flood with a lowered river bed, that is assumed to approximately match the actual one, produces almost the same results in terms of floodplain inundation depths and flow velocities. This shows that the model and the modelling approach is valid and fit for the purpose, i.e. modelling extreme floods and floodplain flow with simplified bed representation as given by the DEM.

In order to show this proof of the model concept we propose to include the additional simulation and the analysis of the results with a lowered river bed in a supplement to the manuscript, and refer to it in the manuscript itself.

[Figure]

*Figure 1: Difference of maximum water depths original simulation minus simulation with lowered river bed.*

[Figure]

*Figure 2 Difference of maximum flow velocities original simulation minus simulation with lowered river bed.*

| model | flooded area [m2] | CSI | Bias [m] (obs. - sim.) | RMSE [m] |
|---|---|---|---|---|
| original DEM 10m | 12369300 | 0.843 | -0.39 | 0.66 |
| modified river bed -0.85m | 12399200 | 0.842 | -0.41 | 0.68 |
| forest roughness n = 0.1 | 12375600 | 0.843 | -0.4 | 0.67 |
| forest roughness n = 0.2 | 12384100 | 0.842 | -0.41 | 0.68 |
| forest roughness n = 0.3 | 12384100 | 0.842 | -0.42 | 0.68 |
| forest roughness n = 0.4 | 12384800 | 0.843 | -0.42 | 0.68 |
| forest roughness n = 0.5 | 12384600 | 0.843 | -0.42 | 0.68 |

About Manning's n, indeed LisFlood has been applied to many situations and locations. However, those other studies do not indicate whether your own study has been applied correctly. You should either do a sensitivity analysis to the roughness values you choose, or use land-use-dependent tabulated values that are generally accepted by the modeling community. Without doing so, there is little reason to have confidence in your result. Again, part of the reason for this is because your value for forest of n=0.05 is very small compared to those listed in Chow (1959) and Arcement & Schneider (1984) of up to n=0.2 for forests and urban areas.

To start the discussion about the roughness parameterization we want to point out that the roughness in simplified hydraulic models has to be seen as an effective parameter, that compensates for model structure simplifications, rather than a "generic and extrapolable physical interpretation" of surface roughness (de Almeida and Bates, 2013). Moreover, the roughness values listed in e.g. Arcement and Schneider (1989) or in the old text books of Ven Te Chow refer to either channel flow,

or to 1D simulations of floodplain inundation. In this context it was also necessary to compensate the model deficiencies in simulating 2D flow over complex terrain by adjusting the roughness. The cited high value of n = 0.2 for urban areas is an example for this, because the flow resistance caused by the urban fabric could not be considered explicitly by the models and had to compensated by a higher roughness. This concept was then extended to (coarse) 2D modelling, in which the urban fabric was also not considered explicitly, but by a higher roughness ("urban porosity"). However, in this study the urban fabric is considered in detail, and the model simulates the flow around buildings on streets. Thus, the low roughness of n = 0.02 for flow over tarmac and concrete was used. Using a high value of n = 0.2 would be contradictory to the modelling approach.

For forests the comparatively low value of n = 0.05 was chosen, because the forests in the area are mainly spruce monocultures with practically no undergrowth. We regarded this type of forest to have a hydraulic roughness not much higher than a normal meadow. Moreover, the forest areas are to a vast extent outside the flooded area, thus not relevant for the inundation simulation. The forest roughness was thus not be seen as critical for the simulation results.

But in order to show this, we conducted additional simulations setting the forest roughness to n = 0.1, 0.2, 0.3, 0.4, and 0.5 respectively. As expected from the rational outlined above, the simulation results hardly changed at all. This is expressed in terms of the practically identical performance measures of the models listed in Table 1, and the histograms of the differences to the original simulation in maximum inundation depths per cell shown in Figure 3. Minor changes occur, because just upstream of the town of Bad Neuenahr-Ahrweiler an area in the floodplain of about 0.15 $km^2$ is classified as forest. These results show that the model reacts to changes in roughness, but in this particular case the question of roughness parameterization of forest does not play a major role. However, in order to avoid similar questions and discussions we will use the simulations with n = 0.2 for forest in the revised manuscript.

[Figure]

*Figure 3: Histograms showing the difference of the simulations using different roughness values n for forest areas, and with a lowered river bed (bottom right) to the original simulation based on the unmodified DEM and forest roughness of n = 0.05.*

About human and vehicle instability, it's probably better that you remove the whole discussion, if you are just going to apply one equation and show the results, as this produces little useful knowledge. Just my opinion, but it seems like useful knowledge could be gained by comparing multiple methods, even if this expands your paper into a full paper rather than a short note.

We disagree in this point. What we want to show with this brief communication is that impact forecasting for a better informed and actionable flood disaster management is possible with the methods and models developed by many researchers in the past years. The map showing the hazard for humans to drown serves as an example, as well as the cited literature for car instability and structural building failure. The purpose is not to discuss the pro's and con's of the different published approaches for these flood impact categories. This was already done in the cited studies on the individual flood impact models, and the use of a particular model has to be decided if these approaches are eventually adopted in operational flood forecasts. The message here is: it is possible, and it adds to a better informed and actionable flood impact forecast.

However, we cite additional studies on human instability in the revised manuscript to show the different studies and models available for this impact category. Citing only a single study is admittedly biased and needs to be corrected. The additional references added are (Jonkman et al., 2008;Milanesi et al., 2015;Penning-Rowsell et al., 2005;REDSCAM, 2000).

**References**

Arcement, G. J., and Schneider, V. R.: Guide for selecting Manning's roughness coefficients for natural channels and flood plains, Report 2339, 1989.

de Almeida, G. A. M., and Bates, P.: Applicability of the local inertial approximation of the shallow water equations to flood modeling, Water Resources Research, 49, 4833-4844, 10.1002/wrcr.20366, 2013.

Jonkman, S., Vrijling, J., and Vrouwenvelder, A.: Methods for the estimation of loss of life due to floods: a literature review and a proposal for a new method, Natural Hazards, 46, 353-389, 2008.

Milanesi, L., Pilotti, M., and Ranzi, R.: A conceptual model of people's vulnerability to floods, Water Resources Research, 51, 182-197, 10.1002/2014WR016172, 2015.

Penning-Rowsell, E., Floyd, P., Ramsbottom, D., and Surendran, S.: Estimating Injury and Loss of Life in Floods: A Deterministic Framework, Natural Hazards, 36, 43-64, 2005.

REDSCAM: The use of physical models in dam-break analysis, Helsinki University of Technology, Helsinki, 56, 2000.